# Inverse Association between Dietary Diversity Score Calculated from the Diet Quality Questionnaire and Psychological Stress in Chinese Adults: A Prospective Study from China Health and Nutrition Survey

**DOI:** 10.3390/nu14163297

**Published:** 2022-08-12

**Authors:** Jia Zhou, Huan Wang, Zhiyong Zou

**Affiliations:** 1The National Clinical Research Center for Mental Disorders & Beijing Key Laboratory of Mental Disorders, Beijing Anding Hospital, Capital Medical University, Beijing 100088, China; 2National Health Commission Key Laboratory of Reproductive Health, Institute of Child and Adolescent Health, School of Public Health, Peking University, Beijing 100191, China

**Keywords:** dietary diversity, psychological stress, diet quality questionnaire

## Abstract

Specific nutrients or dietary patterns influence an individual’s psychological stress. As a major aspect of a healthy diet, the influence of dietary diversity on psychological stress remains uncertain. Within these contexts, we aimed to examine the association between the dietary diversity score and psychological stress, using prospective data from the China Health and Nutrition Survey (CHNS). We included 7434 adult participants, with complete dietary information, in the 2011 wave, and followed-up with perceived stress scale (PSS-14) in the 2015 wave. The dietary intake of foods was coded into 29 food groups, using the DQQ for China, and the dietary diversity scores were obtained, using DQQ, by calculating the number of food groups consumed during one 24-h dietary recall. The univariate analysis, and logistic regression model were used to examine the relationship between psychological stress and diet diversity. Approximately half of the participants (4204, 56.55%) perceived a higher level of stress (PSS-14 total score > 25). Dietary diversity was lower in the higher-stress group (*p* for trend <0.0001). Unconditional multivariate logistic regression demonstrated that participants with higher daily dietary diversity were less likely to experience higher-level psychological stress, compared with participants with lower daily dietary diversity (ORs range: 0.480–0.809). Dietary diversity was found to be inversely associated with psychological stress, in this prospective analysis of a national population. Further studies are required to figure out the mechanism and effectiveness of dietary diversity on psychological stress.

## 1. Introduction

The burden of mental disorders is becoming a worldwide problem [1]. Psychological stress, defined as the sustained, excessive secretion of mental and/or emotional strain from work, family and other daily responsibilities [2], is a specific negative psychological experience that may elicit a host of mental disorders [3]. People living in modern society are faced with multiple stressors, including job stress, financial strain, relationship problems, and adverse life-events. It was estimated that 70% of visits to primary care providers can be attributed to psychological stress [4]. The effects of psychological stress on health require attention.

Several investigations have revealed that healthy dietary patterns, as modifiable lifestyle factors, can help manage stress and prevent stress-related diseases [5,6]. Nutritional psychiatry is an emerging field [7], and experimental evidence on the interplay of nutrition, stress, and mental disorders is increasing [3]. Studies indicate that specific nutrients or foods can influence an individual’s physiological and psychological response to stress [8]. For example, omega-3 polyunsaturated fatty acids and dietary vegetables have been considered to exert stress-buffering effects [9]. High-dose sustained-release ascorbic acid helps to palliate subjective responses to psychological stress [8]. Multivitamin supplementation has a beneficial effect on reducing stress and mood symptoms [10]. Soy lecithin phosphatidic acid and phosphatidylserine complex (PAS) has potential in the treatment of stress-related disorders [11]. A Mediterranean diet supplemented with dairy foods or nuts could exert a beneficial effect on cognitive function and psychological well-being [12,13]. A previous study also demonstrated the significant clinical effect of Mg, B vitamins, rhodiola, and green tea (L-theanine) on relieving stress, after only 14 days of treatment [14]. These studies have focused, more specifically, on the relationship between individual nutrients and psychological stress or stress-related diseases [15]. Despite the role of individual nutrients, a diverse diet is a cornerstone of a sufficient and balanced supply of nutrients [16].

Dietary diversity is a major aspect of a healthy diet [17]. It is defined as the number of foods or food groups consumed individually over a certain period [18]. Although the dietary diversity score cannot perform a comprehensive assessment of nutrient intake, it provides a good assessment of the nutritional adequacy of the diet [19]. The dietary diversity score is a convenient and cost-benefit-integrated indicator of diet quality [20]. In most dietary guidelines, globally, a diverse diet is suggested [21,22], and thought to be one of the best population-engaged approaches to improving public nutrition [23]. According to the results of previous studies, the dietary diversity score is negatively correlated with anxiety [17] and depression [24]. A diverse diet may mitigate cognitive decline and reduce the risk of cognitive impairment in older adults [25]. Moreover, psychological resilience, defined as the ability to cope, adapt, and respond positively to stress, was reported to be positively associated with dietary diversity [26]. However, studies specifically examining the relationship between dietary diversity and psychological stress prospectively among adults are very scarce [27].

Moreover, previous studies calculated dietary diversity using eight food groups [27] or five main food groups [28], which usually depended on quantitative dietary measurement, such as the 24-h dietary recall survey. Recently, the Diet Quality Questionnaire (DQQ) was developed using 29 food groups, to enable the collection of population-level dietary data, based on the framework of global diet quality [29]. The DQQ is a low-burden and standardized method, which has been adapted for the Chinese population, and is used to measure the dietary diversity score [30]. In the present study, we aimed to examine the association between the dietary diversity score, calculated from DQQ for China, and psychological stress, using prospective data from the China Health and Nutrition Survey (CHNS).

## 2. Materials and Methods

### 2.1. Data Resource and Study Participants

In the present study, the data was obtained from the China Health and Nutrition Survey (CHNS), an ongoing, national, household-based cohort study, developed and administered by the Carolina Population Center at the University of North Carolina, and the National Institute of Nutrition at the Chinese Center for Disease Control and Prevention. The CHNS research team employed a multistage, stratified sampling design and included participants from nine provinces to ensure the study’s representativeness. This project was reviewed and approved by the corresponding institutional review committees (2015017). Informed consent was obtained from all participants. Detailed information regarding the project’s description, design, methods and quality control procedures, and the research teams of the CHNS, can be obtained from the cohort profile [31] and the official website (http://www.cpc.unc.edu/projects/china, accessed on 27 May 2022).

Our analysis included the two rounds of survey data collected in 2011 and 2015. There were 15,725 participants in the 2011 wave of the CHNS, and 8737 adult participants were followed up in the 2015 wave. In the 2015 wave, the Perceived Stress Scale (PSS) (Chinese version), was administered in the project for the first time [32], and we included 7434 participants, with information on: basic demographic characteristics (i.e., age, gender, weight (kg), height (m), marital status, province, and urbanization index); complete PSS-14 score; and dietary information (Figure 1).

### 2.2. Study Outcome and Definitions

The perceived psychological stress was measured in 2015 using PSS-14, which was developed by Cohen et al. [33]. It is the most widely-used instrument for measuring psychological stress [32]. The Chinese version of this scale was validated [34]. The PSS-14 comprises 14 items, rated on a 5-point Likert-type scale, ranging from 0: never to 4: very often. Scores are obtained by reverse-scoring the positively stated items (4–7, 9, 10 and 13). Possible scores range from 0 to 56 by summing the scores across all 14 items, with higher scores indicating higher perceived stress. A previous study by Leng et al. (2021) [35] suggested that a total stress score of >25 points can be considered harmful and has a certain degree of negative impact on a person’s physical and mental health. The PSS-14 demonstrated high reliability in our sample (Cronbach α = 0.83).

### 2.3. Dietary Data Collection and Assessment

The quantitative dietary data was collected using three consecutive 24-h recalls, by trained investigators [36]. A previous study had validated the reliability of the 24-h dietary recall [37]. Further details on the dietary interview have been described elsewhere [38].

DQQ is a rapid dietary-assessment tool that captures food-group level data and reflects dietary patterns through sentinel foods (defined as the foods in each food group that were consumed by more than 95% of people) [30,39]. In addition, a previous study had already adapted DQQ for China and verified its reliability in capturing food group consumption in the Chinese population [30]. DQQ, the instrument we used, was designed to obtain and evaluate food-group intake data from only one 24-h recall [30]. Thus, in this study, the dietary intake of foods during the first day only, was coded into 29 food groups using the DQQ for China as follows: (1) staple foods made from grains; (2) whole grains; (3) white roots/tubers; (4) legumes; (5) vitamin-rich orange vegetables; (6) dark-green leafy vegetables; (7) other vegetables; (8) vitamin A-rich fruits; (9) citrus; (10) other fruits; (11) grain-based sweets; (12) other sweets; (13) eggs; (14) cheese; (15) yogurt; (16) processed meats; (17) unprocessed red meat (ruminant); (18) unprocessed red meat (nonruminant); (19) poultry; (20) fish and seafood; (21) nuts and seeds; (22) packaged ultra-processed salty snacks; (23) instant noodles; (24) deep-fried foods; (25) fluid milk; (26) sweetened tea/coffee/milk drinks; (27) fruit juice; (28) sugar-sweetened beverages (SSBs) (sodas); (29) fast food. Dietary diversity scores were obtained using DQQ by calculating the number of food groups consumed during the 24-h dietary recall. More information about DQQ for China was previously described [30].

### 2.4. Measurements and Calculation of Covariates

Information on sociodemographic factors (e.g., age, gender, weight and height, marital status, and urbanization index) was assessed. Body weight and height were measured according to standard procedures. Body mass index (BMI, kg/m^2^) was calculated by the formula: weight (kg)/[height (m)]^2^. Underweight was defined as BMI < 18 kg/m^2^, normal weight was defined as BMI ≥ 18.5 kg/m^2^ and <24 kg/m^2^, overweight was defined as BMI ≥ 24 kg/m^2^ and <28 kg/m^2^, and obese was defined as BMI ≥ 28 kg/m^2^.

The CHNS research team reminded the potential users of the data that the sampling weights are unavailable, and recommended that the users employ control measures, such as the community-level measures of the newly created urbanization index, to control for multilevel, multistage sampling and various multilevel modeling issues [31,40]. Thus, in our study, the urbanization index was controlled as a covariate in multivariate logistic regression, to explore the association between perceived stress and dietary diversity.

### 2.5. Statistical Analysis

Descriptive analyses were used to analyze sample characteristics. The normality of the data distribution was assessed using the Shapiro–Wilk test. Data was summarized in terms of numbers (percentages) for categorical parameters, and medians ± interquartile ranges for continuous parameters that fit a non-normal distribution.

First, univariate analysis was used to analyze the difference, in several variables, between the psychological stress levels. A Wilcoxon rank test was applied for non-Gaussian assumption, to compare differences in continuous parameters between groups (PSS-14 ≤ 25 vs. PSS-14 > 25). For categorical variables, statistical significance between various groups was assessed using Chi-square test. The Cochran–Armitage test was used to examine the trends in dietary diversity across perceived stress-level groups.

Second, we used logistic regression models to explore the association between perceived stress and dietary diversity. Odds ratios (OR) [95% confidence interval] were presented using maximum likelihood methods. Variables adjusted in the model included: age; marital status (never married, married, and divorced/separated/widowed); BMI group (underweight, normal weight, overweight, and obese); gender (female or male); and urbanization index in 2015.

Figures of stratified analyses by gender (female vs. male) were created. The 2-sided *p*-value <0.05 was considered as statistically significant. All analyses were performed using SAS statistical software version 9.4 (SAS Institute Inc., Cary, NC, USA).

## 3. Results

### 3.1. Participant Characteristics

In the analysis, 7434 participants were included. Descriptive statistics of the participants is displayed in Table 1. A total of 4204 (56.55%) people perceived a higher level of stress (PSS-14 total score > 25) and 3230 (43.45%) had a lower level of stress (PSS-14 total score ≤25). Gender, marital status, urbanization index, and BMI showed statistically significant differences between the PSS-14 groups (*p* < 0.05) (Table 1).

### 3.2. The Distribution of Dietary Diversity and Perceived Stress Level

The proportions of different DQQ food groups in the two stress groups are presented in Table 2. Due to the small sample size of participants eating more than 10 species of food in one day, they were aggregated into one group (number of DQQ food groups ≥ 10) in the analysis. According to the result of the Cochran-Armitage test, when compared with the higher-stress group, dietary diversity was higher in the lower-stress group (Z = 7.1100, *p* for trend < 0.0001). In general, the level of psychological stress decreased as the daily dietary diversity increased, for both female and male (Figure 2).

### 3.3. The Relationship between Dietary Diversity and Perceived Stress Level

Unconditional multivariate logistic regression demonstrated that participants with higher dietary diversity in their daily diet were less likely to experience a higher level of psychological stress, compared with participants with lower daily dietary diversity (ORs range: 0.480–0.809) (Table 3).

## 4. Discussion

In this national prospective study using the data from the CHNS, we observed an inverse association between dietary diversity and perceived psychological stress, which remained significant after adjustment for covariates. This finding implies that the more food groups one eats, the less psychological stress one may have.

Diet, as a modifiable environmental factor, plays an important role in modulating psychological stress and preventing stress-related disease [41]. In general support of this idea, there have been recent human and animal studies reporting stress-reducing effects of specific nutrients, or dietary patterns [42]. In animal experiments, calorie restriction, Mediterranean diet, and diets containing prebiotics and/or glycoprotein lactoferrin were proven to reduce psychological stress or enhance stress resilience [43,44,45]. Regarding human research, dietary supplementation with specific nutrients, such as macular carotenoids [42], omega-3 [46], omega-6 [46], Eicosatetraenoic-Acid-enriched phospholipids [47], bioactive components [48], probiotics [49] and B group vitamins [50], was found to reduce psychological distress and mental-disease symptoms, through mechanisms possibly related to neuroinflammation and apoptosis [47]. Moreover, diet pattern was reported to influence psychological distress and stress-induced disorders. For example, Helms, E.R., et al., indicated that high-protein low-fat diet may be effective in mitigating stress and fatigue [51] through alterations in gut microbiota and expression of inflammatory genes [52]. Dietary approaches to stop hypertension, (DASH) dietary patterns, were associated with better mental health in Iranian university students [53]. Additionally, a highly palatable diet, offering a choice of food items, is associated with a reduction in the response to chronic variable stress (CVS) [54]. In addition, a meta-analysis concluded that adhering to a healthy diet and avoiding a pro-inflammatory diet appears to confer some protection against depression [55]. However, no significant associations were observed between a vegetarian dietary pattern and mental-health outcomes in another meta-analysis [56]. Different foods and food groups are good sources of various macro-, and micronutrients [57], so a diversified diet best ensures nutrient adequacy and may play an important role in influencing psychological stress.

In our study, we found that higher dietary diversity was associated with lower psychological stress in Chinese adults, and there was a dose-response relationship. We calculated the dietary diversity score using the data from only one recall. One concern is that it may not represent an individual’s habitual eating patterns as effectively as a calculation using data from more recalls. However, this fact didn’t seem to impact the final result: our study found an association similar to that found by another study, which, using the CHNS 3-day dietary recall of elderly people, found that higher dietary diversity was associated with less psychological stress (ORs range: 0.59–0.63) [27], and even the strengths of association in the two studies were quite similar. Therefore, the number of recalls included in the study might not have fundamentally impacted the finding. This result is also supported by other previous studies investigating the beneficial effect of higher dietary diversity on psychological stress. For instance, dietary diversity was reported to be associated with psychological resilience, which is defined as the ability to cope, adapt, and respond positively to stress [26]. In the longitudinal analysis of Jiang et al., it was demonstrated that the dietary diversity level was negatively associated with the future depression level [58]. Women with higher dietary diversity were reported to obtain lower anxiety scores [17]. In preschoolers, a higher dietary diversity was reported to be associated with a lower likelihood of having mental-health symptoms, such as hyperactivity/inattention, peer relationship problems, and prosocial behavior problems [59]. These all indicate that an enriched diet may have a protective effect against stress or stress-related health issues.

Although the potential mechanism underlying these associations is not clear, there are several possible explanations for how a varied diet protects against psychological stress. First, dietary diversity is a useful proxy of nutritional adequacy and diet quality [18], so higher dietary diversity may be beneficial for mental health. For example, the intake of adequate micronutrients, such as zinc, magnesium, and selenium, is related to the promotion of mental health [60]. Second, previous studies have indicated that an enriched diet is associated with an increased intake of healthy food groups, such as fruits, vegetables, and dairy products [24]. Contrarily, a randomized clinical trial suggested that restricting meat-, fish-, and poultry intake was positively correlated with both dietary diversity and mood state [61]. Therefore, it could be said that with an increase of dietary diversity, the dietary pattern could become more similar to the Mediterranean diet, which is characterized by a high intake of fruits, vegetables, whole grains, legumes, nuts, and seeds and low consumption of red meat [62]. Third, the protective effect of a diverse diet on psychological stress may be attributed to the higher antioxidant intake [63], which has a positive impact on oxidative stress, and immune response [59], protecting the mitochondria and lipids in neuronal circuits [64]. Fourth, higher dietary diversity may also contribute to mental health through a healthier microbiota [27]. Recent findings indicated that decreased dietary diversity corresponded with reduced microbiome diversity [65], which was suggested to be strongly associated with stress-related disorders [66].

## 5. Strengths and Limitations

In this study, we examined the relationship between dietary diversity and psychological stress, prospectively. First, the 29 food group dietary diversity score—according to the China-adapted DQQ—was calculated for the first time for Chinese populations and we observed a higher likelihood of lower psychological stress with higher dietary diversity. This result implies that improving dietary diversity might be an important strategy in reducing psychological stress and improving mental health [59]. Second, this study used the 2011 and 2015 waves of the CHNS data obtained from a national representative sample. This helped to ensure that all relevant types of people were included in the sample, limiting the effect of confounders.

However, several limitations of this study should be noted. First, although covariates, such as age, sex, urbanization index, marital status, and BMI were considered in analyses, other potential confounders were not controlled, such as family history of mental health, physical activity, and income. Second, the cross-sectional nature of the study means that the evidence cannot imply causation. Further animal studies and randomized controlled trials are needed, to validate the interventional effect and causal relationship between dietary diversity and psychological stress. Third, the dietary information was gathered in 2011, and stress was assessed in 2015. While a person’s dietary habits can remain stable over their lifetime, they may also vary with changing circumstances. Therefore, it is a limitation that the diet was not analyzed at both time points. Fourth, while a minimum of three dietary recalls are needed to represent habitual diet, only one of the 24-h diet recalls in 2011 was used to calculate dietary diversity in our study.

## 6. Conclusions

In conclusion, dietary diversity calculated using China-adapted DQQ was inversely associated with psychological stress in this prospective analysis of a national population. Further studies are needed to ascertain the mechanism underlying the association and effectiveness of dietary diversity in reducing psychological stress.

## Figures and Tables

**Figure 1 nutrients-14-03297-f001:**
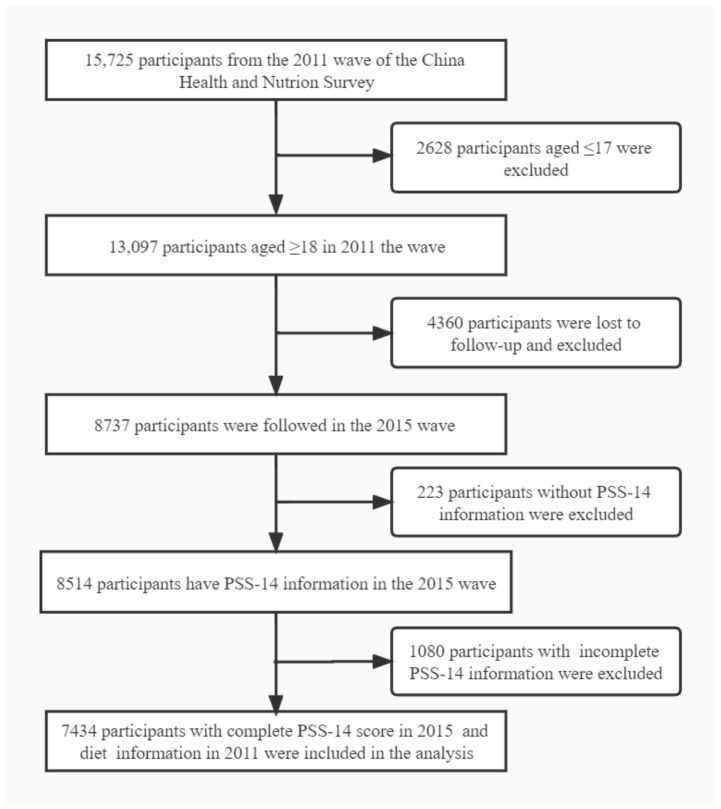
Participant flow diagram.

**Figure 2 nutrients-14-03297-f002:**
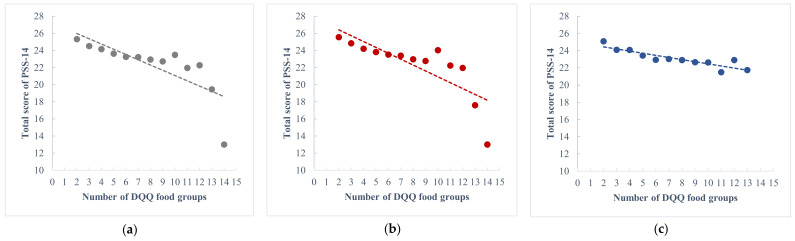
Trends in the relationship between psychological stress and dietary diversity. **(a)** The relationship for all participants; **(b)** The relationship for female participants; **(c)** The relationship for male participants.

**Table 1 nutrients-14-03297-t001:** Descriptive statistics of participants.

Variables *n* (%) or Median (Quartile)	Total Participants (*n* = 7434)	Lower Psychological Stress Group (*n* = 4204)	Higher Psychological Stress Group (*n* = 3230)	Z/ϰ^2^	*p*
Age in 2011	51.00 (41.00–60.00)	51.00 (41.00–60.00)	52.00 (41.00–61.00)	1.1303	0.2584
Age in 2015	55.00 (45.00–65.00)	55.00 (45.00–64.00)	56.00 (45.00–65.00)	1.1444	0.2525
Gender				9.6049	0.0019
Male	3464 (46.60)	2025 (48.17)	1439 (44.55)		
Female	3970 (53.40)	2179 (51.83)	1791 (55.45)		
Marital status in 2011				9.5294	0.0230
Never married	286 (3.86)	171 (4.07)	115 (3.56)		
Married	6530 (88.14)	3712 (88.30)	2818 (87.24)		
Divorced/Separated/Widowed	593 (8.00)	304 (7.23)	289 (8.95)		
Marital status in 2015				11.6203	0.0088
Never married	184 (2.48)	110 (2.62)	74 (2.29)		
Married	6596 (88.97)	3764 (89.53)	2832 (87.68)		
Divorced/Separated/Widowed	634 (8.55)	321 (7.64)	313 (9.69)		
BMI in 2011	23.70 (21.45–26.15)	23.80 (21.49–26.26)	23.61 (21.36–26.03)	−2.6582	0.0079
BMI categories in 2011				13.7777	0.0032
Normal weight	3555 (49.01)	2013 (48.84)	1542 (49.23)		
Obesity	922 (12.71)	545 (13.22)	377 (12.04)		
Overweight	2457 (33.87)	1412 (34.26)	1045 (33.37)		
Underweight	320 (4.41)	152 (3.69)	168 (5.36)		
BMI in 2015	24.09 (21.82–26.52)	24.13 (21.89–26.52)	24.07 (21.75–26.52)	−1.3582	0.1744
BMI categories in 2015				4.8868	0.1803
Normal weight	2986 (44.68)	1709 (44.91)	1277 (44.37)		
Obesity	968 (14.48)	571 (15.01)	397 (13.79)		
Overweight	2454 (36.72)	1382 (36.32)	1072 (37.25)		
Underweight	275 (4.11)	143 (3.76)	132 (4.59)		
Residence				53.6550	<0.0001
Rural	4563 (61.38)	2428 (57.75)	2135 (66.10)		
Urban	2871 (38.62)	1776 (42.25)	1095 (33.90)		
Province				229.9176	<0.0001
Beijing	702 (9.44)	413 (9.82)	289 (8.95)		
Chongqing	548 (7.37)	240 (5.71)	308 (9.54)		
Guangxi	606 (8.15)	293 (6.97)	313 (9.69)		
Guizhou	613 (8.25)	296 (7.04)	317 (9.81)		
Heilongjiang	518 (6.97)	294 (6.99)	224 (6.93)		
Henan	602 (8.1)	316 (7.52)	286 (8.85)		
Hubei	610 (8.21)	314 (7.47)	296 (9.16)		
Hunan	561 (7.55)	304 (7.23)	257 (7.96)		
Jiangsu	759 (10.21)	493 (11.73)	266 (8.24)		
Liaoning	608 (8.18)	460 (10.94)	148 (4.58)		
Shandong	569 (7.65)	292 (6.95)	277 (8.58)		
Shanghai	738 (9.93)	489 (11.63)	249 (7.71)		
Urbanization index in 2011	73.84 (54.55–88.93)	77.18 (56.16–89.77)	68.92 (53.08–87.88)	−8.0879	<0.0001
Weight in 2011, kg	61.40 (54.50–70.00)	62.25 (55.00–70.20)	60.40 (53.50–68.60)	−6.4470	<0.0001
Height in 2011, cm	161.00 (155.50–167.80)	162.00 (156.00–168.00)	160.00 (154.90–166.60)	−7.0626	<0.0001
WC in 2011, cm	84.00 (77.00–91.10)	84.50 (77.00–91.70)	83.20 (76.50–90.60)	−3.1637	0.0016
Urbanization index in 2015	77.08 (60.12–87.60)	79.74 (61.02–88.55)	73.57 (57.97–86.98)	−7.9719	<0.0001
Weight in 2015, kg	62.30 (55.00–70.40)	62.90 (55.70–71.00)	61.60 (54.20–69.80)	−4.8665	<0.0001
Height in 2015, cm	160.70 (155.00–167.10)	161.50 (155.80–168.00)	160.00 (154.00–166.00)	−6.5127	<0.0001
WC in 2015, cm	85.00 (78.00–92.50)	85.60 (78.50–93.00)	85.00 (77.35–92.00)	−2.7671	0.0057
Cumulative average dietary intake					
Energy, kcal/day	1835.11 (1407.41–2359.90)	1841.62 (1407.83–2385.35)	1820.19 (1407.40–2329.92)	−1.0028	0.3159
Protein, g/day	63.04 (46.23–84.70)	63.99 (46.85–86.02)	61.62 (45.56–82.87)	−3.7050	0.0002
Carbohydrate, g/day	255.30 (184.39–342.15)	255.12 (181.50–340.87)	255.91 (186.87–344.86)	1.2833	0.1994
Fat, g/day	58.90 (35.97–87.89)	59.85 (37.24–89.23)	58.05 (34.05–86.01)	−2.9658	0.0030
Calcium, mg/day	363.27 (241.08–539.65)	376.55 (248.32–561.32)	345.63 (232.72–512.31)	−5.4703	<0.0001
Sodium, mg/day	3761.39 (2655.46–5272.83)	3781.94 (2662.10–5307.16)	3740.80 (2648.67–5230.20)	−0.6063	0.5443

Note: BMI, body mass index; WC, waist circumference. Continuous variables are expressed as median (quartile). Categorical variables are expressed as numbers (percentages).

**Table 2 nutrients-14-03297-t002:** Dietary diversity and psychological stress level *n* (%).

Variables	Lower Psychological Stress Group	Higher Psychological Stress Group	Z	*p*
Dietary diversity (Number of DQQ food groups)			7.1100	<0.0001
2	51 (1.21)	68 (2.11)		
3	268 (6.37)	290 (8.98)		
4	580 (13.80)	534 (16.53)		
5	822 (19.55)	650 (20.12)		
6	924 (21.98)	658 (20.37)		
7	722 (17.17)	516 (15.98)		
8	430 (10.23)	286 (8.85)		
9	242 (5.76)	130 (4.02)		
≥10	165 (3.92)	98 (3.03)		
10	86 (2.05)	63 (1.95)		
11	51 (1.21)	23 (0.71)		
12	20 (0.48)	10 (0.31)		
13	7 (0.17)	2 (0.06)		
14	1 (0.02)	0 (0.00)		

**Table 3 nutrients-14-03297-t003:** Associations of dietary diversity with psychological stress.

Parameter	df	Estimate	Standard Error	Wald ϰ^2^	*p*	OR	95%CI
Intercept	1	0.9343	0.2950	10.0287	0.0015		
BMI categories in 2015							
Normal weight	Ref						
Obesity	1	−0.0679	0.0757	0.8035	0.3701	0.934	0.806, 1.084
Overweight	1	0.0586	0.0555	1.1128	0.2915	1.060	0.951, 1.182
Underweight	1	0.1769	0.1274	1.9280	0.1650	1.194	0.930, 1.532
Gender							
Female	Ref						
Male	1	−0.1579	0.0507	9.6983	0.0018	0.854	0.773, 0.943
Marital status in 2015							
Never married	Ref						
Married	1	−0.1336	0.1784	0.5604	0.4541	0.875	0.617, 1.241
Divorced/Separated/Widowed	1	0.000463	0.2044	0.0000	0.9982	1.000	0.670, 1.493
Age	1	0.000727	0.00201	0.1314	0.7170	1.001	0.997, 1.005
Urbanization index	1	−0.00836	0.00160	27.3869	<0.0001	0.992	0.989, 0.995
Dietary diversity (Number of DQQ food groups)							
≤2	Ref						
3	1	−0.2118	0.2251	0.8850	0.3468	0.809	0.521, 1.258
4	1	−0.3943	0.2155	3.3469	0.0673	0.674	0.442, 1.029
5	1	−0.4587	0.2143	4.5814	0.0323	0.632	0.415, 0.962
6	1	−0.5350	0.2149	6.1944	0.0128	0.586	0.384, 0.893
7	1	−0.5036	0.2170	5.3890	0.0203	0.604	0.395, 0.925
8	1	−0.4899	0.2241	4.7768	0.0288	0.613	0.395, 0.951
9	1	−0.7331	0.2404	9.3010	0.0023	0.480	0.300, 0.770
≥10	1	−0.6858	0.2513	7.4484	0.0063	0.504	0.308, 0.824

Note: 761 observations were deleted due to missing values for the explanatory variables.

## Data Availability

The dataset in the present study was open-accessed and freely obtained from the CHNS website with registration at https://www.cpc.unc.edu/projects/china/data/datasets/ (accessed on 22 March 2021).

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
