# Peer review of "Inverse Association between Dietary Diversity Score Calculated from the Diet Quality Questionnaire and Psychological Stress in Chinese Adults: A Prospective Study from China Health and Nutrition Survey"

_nutrients, 2022, doi:10.3390/nu14163297_

Round 1
Reviewer 1 Report
The authors present an interesting paper exploring the association between dietary diversity and stress in Chinese adults. The findings from this provide novel insights into the association between dietary diversity and stress. The authors apply rigorous analyses to explore the associations between dietary diversity and stress using both continuous and categorical data. The findings can be used to help guide the development of clinical trials and interventions targeting modifiable lifestyle factors for mental health. However, there are grammatical errors and other issues that make it difficult to follow the core content of the manuscript. The manuscript needs to be refined and edited. Below are my suggestions.
Abstract
· Grammatical errors need to be corrected e.g., line 12 should be ‘specific nutrients or dietary patterns…’
· The methods used to calculate dietary diversity are unclear (lines 17-18).
· When referring to a higher intake of DQQ food groups per day are you referring to greater dietary diversity? Make sure this is kept consistent throughout the manuscript.
Introduction
· Grammatical errors need to be corrected and overall flow needs to be improved. Some of the errors found will be listed in additional minor comments.
· In the review of past studies, it may be worth including some of the evidence for whole dietary patterns such as the Mediterranean diet and B-vitamins.
· I think it would be beneficial to include an aim at the end of the introduction.
Methods
· Correcting the grammatical errors and improving the sentence structure will help the methods to become clearer.
· Please include more details about the study cohort or refer to a paper that describes this in more detail. For example, describe when data was collected (this is currently only described in the abstract), was ethics approval required for this research? Did participants provide informed consent to participate?
· Are the dietary methods used validated (e.g., 24 hrs dietary recall used and the DQQ)?
· It is unclear how the dietary data gathered in the 24hr diet recall was used to calculate the DQQ. More detail is needed. Where the authors state that ‘the dietary intake of foods during the first day were coded into 29 food groups’ does this mean that only one of the 24-hr diet recalls was used. If so, this should be discussed in the limitations as a minimum of three dietary recalls is needed to represent habitual diet.
· If the continuous variables were not normally distributed, were the assumptions for the regressions met?
Results
· The results can be improved through editing to correct grammatical errors and improve flow
· Table 1: it is unclear what is being reported in the lower group and higher group columns. Are these means and ranges? This should be described in the table. The lower group and the higher group should also be more specific, e.g., lower stress
· Table one should also include a column including the statistics for the whole sample.
Discussion
· The overall flow and readability of the discussion can be improved through editing.
· It is unnecessary to repeat the statistical findings in the discussion if they have been reported in the results (line 221)
· It is unclear whether you are talking about the strengths of the study in paragraph five. This should be clearly stated, otherwise it just sounds like a repetition of what was done.
· The limitations of this study need to be expanded upon. Firstly, the cross-sectional nature of the study means that you cannot imply causation. This needs to be clearly stated in the discussion. In addition, the timing of the assessments is a limitation that needs discussion, diet was assessed in 2011 and stress was assessed in 2015, while a person’s dietary habits can remain stable over their life, they can also vary with changing circumstances. Therefore, it is a limitation that the diet was not assessed at both time points.
Additional minor comments:
· Line 14: what context are the authors referring to?
· Should not start a sentence with a number e.g., 4300. Should instead spell the number out -Four-thousand-three hundred. Lines 20 and 146.
· Line 35: Add ‘The’ to the start of the sentence.
· Line 48: it may be better to refer to Omega 3 oils as omega-3 polyunsaturated fatty acids. Dietary vegetable should be dietary vegetables
· Line 56: ‘of health diet’ should be ‘of a healthy diet’
· Lines 70-71: remove ‘the’.
· Line 73: Replace of with using in ‘was developed of 29 food groups’
· Line 146: Basic information should be changed to descriptive statistics
· Line 154: The meaning of this sentence is unclear ‘The prevalence of number of DQQ food groups in stress group was presented in table 2’
· Line 199: ‘specific nutrient or dietary pattern’ should be plural e.g., specific nutrients or dietary patterns
Author Response
Thank you very much for your positive and helpful comments. We have adopted all the suggestions and the manuscript has been thoroughly revised and edited by a native speaker. In the attached file are our point-by-point responses to each of the comments.

Reviewer 2 Report
I would like to thank the author(s) for your submission and appreciate the opportunity to read and review your manuscript. This study adds to the literature on the association between dietary diversity and psychological stress in Chinese adults. It is a nice manuscript that addresses an important topic.
Review comments
- Methods
- No research design for this study was reported. In addition, descriptions of the original datasets and the sample are missing.
- It isn’t clear which variables were selected from which datasets (2011 or 2015).
- 2.2 Study outcome and other definitions => what are other definitions?
- Statistical Analysis
- The association between perceived stress and dietary diversity was examined in logistic regression models, controlling for sociodemographic variables. Conducting multiple linear regression to assess the relationship between diet diversity and perceived stress seems redundant. Furthermore, non-parametric statistics were used as the variables were not normally distributed, so performing multiple linear regression doesn’t make sense as the normality assumption is violated.
- Authors already conducted the Cochran-Armitage test to examine trends in dietary diversity across perceived stress level groups. Spearman’s correlation between perceived stress and dietary diversity does not seem necessary.
- No information about the use of the weights was reported. Please clarify it.
- Results
- The author(s) used diverse terms to explain each variable throughout the manuscript, for example, the number of DQQ food groups => dietary diversity; PSS-14 total score => perceived stress. I would recommend using terms consistently to let the audience understand easily.
- Table 2 and figure 2 represent the same results. Please delete either one.
- Figure S1 also seems redundant and not necessary.
- Discussion
- Discussion needs to reflect on data findings without drifting and making leaps in conclusions that do not reflect or pertain to the current data findings. For example, on Page 10, lines 193-195, the current analysis findings do not seem to support the leap that it is significant in the primary prevention of mental illness.
Author Response
Thank you very much for your appreciation and insightful comments. All your comments are invaluable and helpful for improving our paper. In the attached file are our point-by-point responses to each of the comments.

Round 2
Reviewer 1 Report
The authors have made substantial revisions to this manuscript. As a result, the paper reads much better, and the authors have addressed most of my concerns.
Below are some additional comments:
Abstract
Line 19: Clarify the number of 24hr diet recalls used to calculate diversity scores
Introduction:
The readability and flow of the introduction has greatly improved. Two typos need to be corrected. Line 54: psychilogical should be corrected to with psychological. Line 60: an should be a.
Materials and Methods:
Please clarify lines 87-93. Potentially move the discussion of the use of urbanisation index o the section-Measurements and calculation of covariates
Please clarify how many 24hr diet recalls were used to calculate the dietary diversity scores. In the discussion, it states that only one recall was used. Would it be possible to include an average from the multiple recalls to represent participants' habitual diets?
Line 122: Please clarify what you mean by sentinel
Results:
Please clarify how the cut-offs for high and low stress (PSS-14) were chosen, do these cut off have any clinical significance?
Table 1 still needs better labelling. Unclear what the number in column two are
The authors need to describe figure 2 better. Unclear how the first two graphs differ from each other.
Discussion:
Reduce the discussion of animal studies and nutrients and focus more on the existing evidence for dietary diversity and whole dietary patterns.
If only one diet recall is used, make sure that this is specifically discussed when interpreting the results. For example, the dietary diversity score only captures the diversity in one day and may therefore not represent someone's habitual eating patterns. How does this impact the interpretation of the results?
Author Response
Thank you very much for your insightful comments. All your comments are helpful for improving our paper. In the attached file are our point-by-point responses to each of the comments.
